# Sources and Determinants of Wholegrain Intake in a Cohort of Australian Children Aged 12–14 Months

**DOI:** 10.3390/ijerph17249229

**Published:** 2020-12-10

**Authors:** Amira Hassan, Gemma Devenish, Rebecca K. Golley, Diep Ha, Loc G. Do, Jane A. Scott

**Affiliations:** 1School of Public Health, Curtin University, Perth 6012, Australia; amira.hassan@postgrad.curtin.edu.au (A.H.); jane.scott@curtin.edu.au (J.A.S.); 2Caring Futures Institute, College of Nursing and Health Sciences, Flinders University, Adelaide 5001, Australia; rebecca.golley@flinders.edu.au; 3Australian Research Centre for Population Oral Health, School of Dentistry, The University of Adelaide, Adelaide 5000, Australia; diep.ha@adelaide.edu.au (D.H.); loc.do@adelaide.edu.au (L.G.D.)

**Keywords:** wholegrains, socio‑economic determinants, dietary assessment, 24‑hour recall, food record, food sources, early childhood, complementary feeding

## Abstract

In the first two years of life, exposure to wholegrain foods may help establish life‑long consumption patterns associated with reduced risk of chronic disease, yet intake data are lacking for this age group. This cross‑sectional analysis aimed to determine intakes and food sources of wholegrains in a cohort of 828 Australian children aged 12–14 months, and to identify determinants of wholegrain intake. Three non‑consecutive days of dietary intake data were collected using a 24‑h recall and 2‑day estimated food record. The multiple source method was used to estimate usual wholegrain intake, and the multivariable general linear model procedure used to identify associations between usual wholegrain intake and socio‑demographic determinants. The mean wholegrain intake was 19.5 (±14) g/day, and the major food sources were ready to eat breakfast cereals (40%) breads and bread rolls (26.6%), flours and other cereal grains (9.4%), and commercial infant foods (8.3%). Lower wholegrain intakes were observed for children whose mothers were born in China (*p* < 0.001) and other Asian countries (*p* < 0.001), with the exception of India (*p* = 0.193); those with mothers aged less than 25 years (*p* = 0.001) and those with two or more siblings (*p* = 0.013). This study adds to the weight of global evidence highlighting the need to increase children’s intake of foods high in wholegrain, including in the first few years of life.

## 1. Introduction

A growing body of evidence indicates that a diet rich in wholegrains may reduce the risk of cardiovascular disease, diabetes, and colorectal cancer [1,2,3,4]. There is evidence also that food preferences and eating behaviors are established in the first few years of life and track into adulthood [5], and that nutrition in early life is linked to disease outcomes in adulthood [6,7]. Therefore, it is important that children are exposed to a variety of wholegrain foods in these early years, while lifelong eating habits are being formed.

However, information on the wholegrain intakes of Australian children under the age of 2 years is scarce. While a number of international studies [8,9,10,11,12,13], and one Australian study [14], have reported the wholegrain intake of children, only one of these included children aged less than 2 years, for whom results were aggregated with children aged 1.5–5 years [11]. These studies all found that most pre-school and school‑aged children consumed inadequate amounts of wholegrain [10,11,12,13] and several studies identified a positive correlation between wholegrain intake and overall diet quality [8,9,10]. Neither the most recent 2011–2013 Australian Health Survey [15] nor the 2007 National Children’s Nutrition and Physical Activity Survey [16] examined the dietary intakes of children younger than 2 years of age, so dietary reporting of this group in Australia tends to come from longitudinal cohort data. The Study of Mothers’ and Infants’ Life Events affecting oral health (SMILE) is one of a few such studies providing data on this age group [17,18,19,20,21,22,23,24,25].

Therefore the aims of this study are to describe the intake and food sources of wholegrain for a cohort of Australian children aged between 12 and 14 months, and investigate socio‑demographic predictors of wholegrain consumption.

## 2. Materials and Methods

### 2.1. Data and Study Population

This study is a secondary analysis of data from the SMILE birth cohort, consisting of 2181 children born to 2147 mothers recruited between July 2013 and August 2014 from three major maternity hospitals in Adelaide, South Australia [22,26]. Mothers who had a sufficient understanding of the English language and were not intending to move from Adelaide for at least one year following recruitment were eligible to participate in the study. Participants from hospitals in areas of low socioeconomic status were oversampled to compensate for anticipated higher attrition rates from these areas [22]. Signed, informed consent was obtained from all mothers. Socio‑demographic characteristics were collected via a paper-based questionnaire at recruitment, and the participants’ choice of postal, online or telephone‑administered questionnaire when the children were 3, 6, and 12 months of age. The dietary data used in this analysis was collected at around 12 months of age, however SMILE will continue to follow the cohort until at least 2022. This study was approved by the Southern Adelaide Clinical Human Research Ethics Committee (HREC/50.13, approval date: 28 February 2013) and the South Australian Women and Children Health Network (HREC/13/WCHN/69, approval date: 7 August 2013).

### 2.2. Dietary Analysis

The methods used to collect the 12-month dietary data used in this analysis have been described in detail previously [21]. Briefly, 3 days of dietary intake data, comprising 2 weekdays and 1 weekend day, were collected via a single telephone‑based 24‑h recall and 2 non‑consecutive days of estimated food records. The data were entered into FoodWorks version 9 (Xyris Software (Australia) Pty Ltd., Brisbane, Australia) for analysis using the AUSNUT 2011‑13 Food, Supplement and Nutrient Database [27]. Trained nutritionists/dietitians conducted the phone interviews and double‑entered all dietary data, following protocols for both the survey and data entry activities and employing calibration methods to ensure standardization between researchers [21].

Wholegrain intake was determined using an AUSNUT 2011‑13 compatible wholegrain database produced by Galea et al. [28]. Due to the limited number of commercial infant and toddler foods in the AUSNUT 2011‑13 database and the rapidly growing market, 187 new foods were added to the database for analysis [21]. Wholegrain values were determined for these items following the procedure for wholegrain content calculation established by Galea et al. (Figure 1) [28], using information from product nutrition information panels and ingredients lists.

### 2.3. Statistical Analysis

Dietary data were exported from FoodWorks and merged with the wholegrain values determined for the infant foods using Microsoft Access (Microsoft Office 2016, Albuquerque, NM, USA). Intra‑individual variability of consumption was addressed using the EFCOVAL Consortium Multiple Source Method (MSM) (Nuthetal, Germany) [29]. The MSM used the wholegrain intake data obtained from all three days of dietary assessment to estimate usual wholegrain intake in grams per day (g/day) for each child. Data were then imported into SPSS version 25.0 (IBM Corporation, Armonk, NY, USA) for statistical analysis. The median, interquartile range, mean, standard deviation and 95% confidence interval (CI) of usual wholegrain intake were calculated for the whole sample and by strata of socio‑demographic characteristics.

Food sources of wholegrain were identified by grouping all foods consumed into major and sub‑major food groups using the AUSNUT 2011‑13 food group coding system [27]. The percentage contribution of each food group to total wholegrain intake was determined, and mean wholegrain intake values for each food group were calculated for all children and for consumers only.

Explanatory variables collected at baseline and investigated as potential predictors of wholegrain intake included maternal age (<25 years, 25–34 years, and ≥35 years), maternal level of education (high school/vocational and some university or above), total number of children (1, 2, and ≥ 3), child sex (male and female), maternal country of birth (Australia and New Zealand, United Kingdom (UK), India, China, Asia—other, and Other) and Index of Relative Socio‑economic Advantage and Disadvantage (IRSAD). Residential postcodes were used to obtain a household‑level measure of socio‑economic status using the IRSAD deciles, where 1 equals the most disadvantaged and 10 the most advantaged [30]. This Index is developed by the Australian Bureau of Statistics based on data from the five-yearly census [30]. For the purpose of this investigation, the IRSAD deciles were collapsed into five pairs (deciles 1–2, deciles 3–4, deciles 5–6, deciles 7–8, and deciles 9–10). Age of introduction of complementary food (solids) (<17 weeks and ≥17 weeks) was derived from information collected at 3, 6, and 12 months, and was also investigated as a potential predictor of wholegrain intake.

The independent associations between usual wholegrain intake (g/day), and the identified explanatory variables were investigated simultaneously using the multivariable general linear model (GLM) procedure. The model adjusted for child age (months) at the time the 24-h recall was taken, to account for age-related variations in the total volume of food consumed. Although the outcome variable was slightly positively skewed, the standardized residuals from the multivariable regression model did not exhibit any large departures from normality. Results are presented as the unadjusted and adjusted mean usual wholegrain intake, with 95% CI and the regression coefficient, standard error and *p*-values obtained from regression analyses. For all statistical analyses, a *p*-value of <0.05 was considered statistically significant.

### 2.4. Sensitivity Analysis

Sensitivity analysis was conducted to account for extreme over‑ or under‑reporting of energy intakes (EI) [31]. A reference value for estimated energy requirement (EER) [32] was assigned to each participant based on their sex and age at time of 24‑h recall. Participants were said to have plausible dietary intakes if they had a ratio of EI:EER between 0.54 and 1.46 [33]. To assess the robustness of the findings, primary analyses were conducted on the entire sample and then repeated with only those participants with plausible energy intakes.

## 3. Results

### 3.1. Participant Characteristics

A total of 1165 mothers completed the 24‑h recall interview and 847 returned the food record. Complete dietary data for all 3 days were available for 828 children. Previous investigations of population parameters have found that the sample characteristics of this cohort are diverse and generally representative of the total South Australian Births that year [19,21,26]. The majority of participating mothers were aged between 25 and 34 years of age, had commenced or completed university studies, were born in Australia or New Zealand, and had one or two children (Table 1). Socio‑economic position measured by IRSAD was relatively evenly spread over the five categories. The mean age of the children included in this analysis was 13.1 (±0.8) months (median 12.9 months, IQR 12.6–13.3 months), with most children (92%) aged between 12 and 14 months.

### 3.2. Wholegrain Intake and Sources

The mean usual wholegrain intake for all participants was 19.5 g/day (95% CI 18.5–20.4) (Table 1). The largest contribution to wholegrain intake was provided by ready‑to‑eat breakfast cereals (40.0%), followed by breads and bread rolls (26.6%), flours and other cereal grains and starches (9.4%), and commercial infant foods (8.3%) (Table 2).

### 3.3. Determinants of Wholegrain Intake

Lower wholegrain intakes were observed for children whose mothers were born in China (*p* < 0.001) and other Asian countries (*p* < 0.001), with the exception of India (*p* = 0.172), than for those whose mothers were born in Australia or New Zealand (Table 3). Additionally, children whose mothers were less than 25 years of age (*p* = 0.010) and those with two or more siblings (*p* = 0.016) had lower intakes than those born to mothers aged 35 years and older, and those with no siblings. There were also lower wholegrain intakes among girls (*p* = 0.025) than boys, however the unadjusted mean usual wholegrain intakes of these two groups differed by just 1.4 g.

### 3.4. Sensitivity Analysis

Removal of 125 participants with implausible energy intakes resulted in findings similar to the primary analysis (Appendix A), with mother’s country of birth (China and other Asian countries, *p* < 0.001), young maternal age (*p* = 0.006), and female sex (*p* = 0.032) being independently associated with lower wholegrain intakes. While nonsignificant (*p* = 0.64), children with two or more siblings still had lower intakes than those without siblings.

## 4. Discussion

This study reports on the food sources and predictors of wholegrain intake for an Australian cohort of children aged 12–14 months. Globally, few countries have targets for the intake of wholegrains, however those that do recommend that most, or at least half, of all grain products consumed be wholegrain [34]. Quantitative recommendations, particularly those for children, are scarce. The American Heart Association recommends that children aged 1 year should consume 1oz (28.4g) and children aged 2 to 3 years should consume 1.5oz (~42.5 g) of wholegrain per day [35]. The Australian Grains and Legumes Nutrition Council (GLNC) recommends a daily wholegrain target intake of 24 g for children aged 1 to 3 years [36]. The mean usual wholegrain intake for the SMILE cohort was 19.5 g, and less than one third (29.3%) of children met the GLNC target of 24 g.

Direct comparisons of wholegrain intakes between studies internationally is problematic as the age groupings of children varies and often includes older children. Older children would likely be consuming larger quantities of food and therefore have correspondingly higher intakes of wholegrain. Additionally, varying methods are used to define, quantify and assess wholegrain intakes across studies, and wholegrain intake is reported as either the mean or median, further complicating comparisons between studies. In Australia, the median reported intake of wholegrain was 19.1 g/day for children aged between 2 and 3 years in a secondary analysis of national survey data [14]. International studies have reported a wide range of whole grain intakes including older children from the USA (2–8 years, 9 g/day) [12], Ireland (5–10 years, 18.5 g/day) [10], France (3–6 years, 3 g/day) [8], Italy (>3 years, 2 g/day) [9], Malaysia (7–12 years, 2.2 g/day) [13], and UK (1.5–5 years, 27.3 g/day) [11]. The present study adds to the weight of global evidence highlighting the need to increase children’s intake of wholegrains, emphasizing this importance in the first few years of life.

These inconsistencies in reporting between studies highlight the importance of having wholegrain targets for age. A set of quantitative targets for Australian children and adults should be considered for inclusion in the next review of the Australian Dietary Guidelines, to supplement the current guidance to choose mostly wholegrain foods within the grains group [37]. This would highlight to parents and practitioners the importance of including wholegrain foods in their child’s diet, and serve as a practical guide for them to follow. Impacts would extend further, with the target serving as a mandate for action by policymakers, the food industry and food service providers. Such a quantitative wholegrain recommendation will also aid future researchers in better evaluating wholegrain intake within this age group.

Cereals and cereal products, specifically ready‑to‑eat breakfast cereals and regular breads and bread rolls, were the highest contributors to wholegrain intakes of children in this study, similar to findings in older Australian children [14] and other international studies [8,9,10,11,12,13]. This is not unexpected, as cereals and breads are staple foods in the Australian diet [14,15,28] reflected in the affordability, accessibility and convenience of these food items. There is scope for parents, caregivers and early childhood food service providers, such as daycare centers, to increase the range of foods which contribute to children’s wholegrain intake. For example, introducing brown rice and other wholegrains, wholemeal flour, and wholegrain bread in the first years of life may support a taste preference for foods rich in wholegrain.

While cereal products may be a major source of wholegrains, many are also a major source of free sugars. We have previously reported that cereals, cereal products and cereal‑based dishes contributed 24.5% of free sugars intake in this cohort [21]. A similar finding was observed in a number of international studies with high consumption of ready‑to‑eat breakfast cereals being linked to high sugars intakes [8,9,11]. Among children and adults, diets high in free sugars have been linked to poor overall diet quality, obesity, dental caries, and other non‑communicable diseases [38]. As a key dietary staple, cereals and cereal products are ideal foods to target for a national product reformulation scheme, to promote lower intakes of free sugars in children and adults. Further, neither wholegrains nor free sugars are currently mandatory in Australian food labelling via nutrition information panels [39], and they are not included in the algorithm for the national front of pack labelling scheme [40]. Clearer and more transparent labelling requirements for these nutrients would empower consumer choice and encourage product reformulation, contributing to improved diet quality and health outcomes [41,42].

Numerous studies have reported that the diets of children born to mothers who were younger and with lower education levels are lower quality than the diets of children born to older and highly-educated mothers [43,44,45,46,47]. While we found an inverse association between wholegrain intake and maternal age, we did not find an independent direct association with level of maternal education, nor household socio-economic position (IRSAD).

In this study, children with two or more siblings were found to have significantly lower wholegrain intakes than those with no siblings. This finding is consistent with other Australian [48] and international studies [7,45,49] which have reported poorer diet quality among higher birth order children. This is likely a result of financial and time constraints associated with larger families, making it difficult for caregivers to consistently provide nutrient‑dense meals [45]. Additionally, it has been suggested that less stringent parenting practices are employed in multiple‑child families when compared to single‑child families [50] and that a child’s food exposures are shaped by the food preferences of older siblings [45,51]. These factors may lead to a greater consumption of energy‑dense, nutrient‑poor, non‑core foods and drinks within multiple‑child families [19,52], which may displace foods high in wholegrain.

Children whose mothers were born in China and other Asian countries, with the exception of India, were found to have significantly lower wholegrain intakes compared to those whose mothers were born in Australia or New Zealand. This may be a consequence of maternal adherence to traditional dietary patterns, as non‑wholegrain based carbohydrates, like white rice and white‑flour noodles, are common food staples in many Asian countries [1,13]. Children whose mothers were born in India may have presented with higher intakes of wholegrain relative to those whose mothers were born in other Asian countries due to the adoption of a more western diet pattern upon migration, or because Indian dietary patterns are high in total carbohydrate, including both refined and wholegrain forms [53].

Lower usual wholegrain intakes were observed in female children than male children, although this difference was relatively small. This difference may be explained by variations in total energy intake between these two groups, which was not adjusted for in the multivariable analyses. A comparison of wholegrain intake of British male and female children aged 1.5 to 17 years reported a loss of significance when intakes were adjusted for energy [11].

A strength of the present study was the intentional oversampling of mothers and infants from socio‑economically disadvantaged areas to compensate for predicted high rates of attrition [22]. This resulted in a study population that was socioeconomically diverse and generally representative of the population from which they were derived [26]. A further strength of this study was the dietary data collection methods. We used a 24-h recall and 2-day food record to obtain dietary data for three non‑consecutive days, which allowed us to estimate usual wholegrain intake and to adjust for intra‑individual variability. A limitation of this study is that this dietary data, collected in 2014–2015, may not reflect current dietary patterns of infants in light of the rapidly changing food supply, particularly the commercial infant and toddler food market [54]. A further limitation was the use of parental‑proxy reporting, which is susceptible to differing forms of misreporting, including social desirability biases [55]. Additionally, parents may find it difficult to quantitatively estimate food intakes of young children, as they typically consume small portions of food and generate high levels of plate waste [56], which may exacerbate errors associated with misreporting.

## 5. Conclusions

While Australia lacks an official dietary target, the mean wholegrain intake of this cohort of Australian 12–14-month-olds was below recommendations from the (non-government) Australian Grains and Legumes Nutrition Council. This study is the first to report wholegrain intakes in Australian children under 2 years of age, and adds to the weight of global evidence highlighting the need to increase children’s intake of wholegrains, including in the first few years of life. Parents, in particular younger mothers, those with larger families, and those from Asian countries, need practical guidance on the sources and amounts of wholegrain products that their children should be eating. This guidance should be supported by the actions of policymakers and food manufacturers, through greater transparency in food labeling and product reformulation schemes. The adoption of national or international quantitative wholegrain intake targets by age will assist future research, policymaking and practice.

## Figures and Tables

**Figure 1 ijerph-17-09229-f001:**
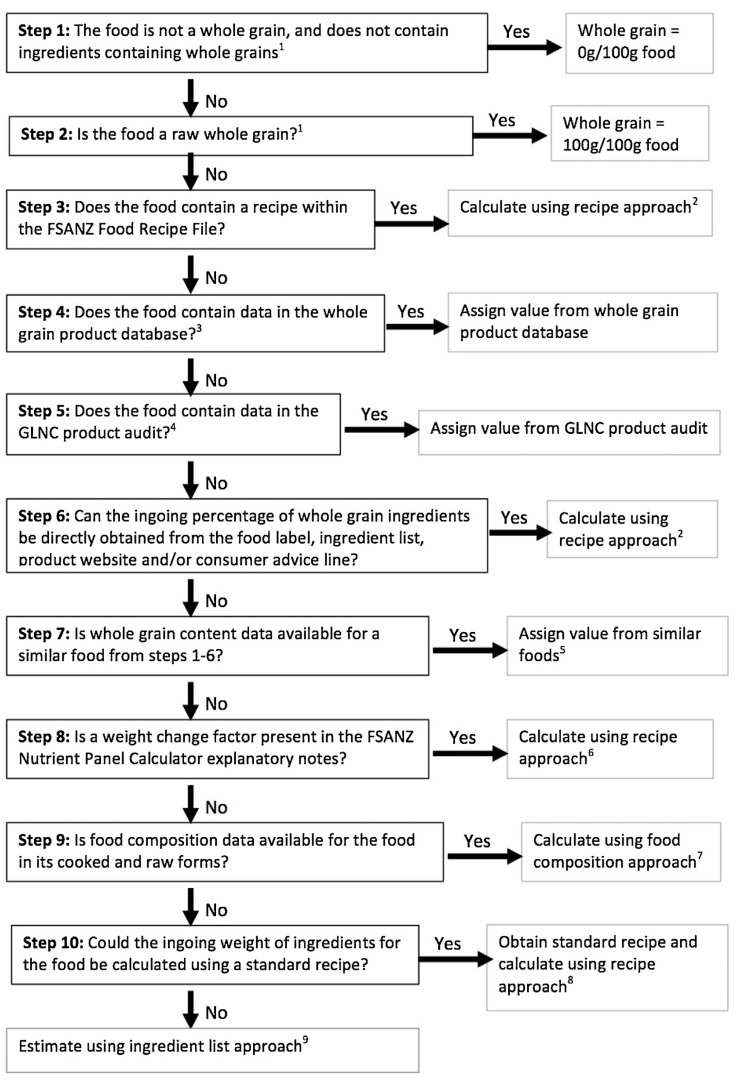
Schematic process for calculation of wholegrain content of foods within the AUSNUT 2011‑13 Food, Supplement, and Nutrient Database [28]. Reprinted from The Journal of Food Composition and Analysis, Vol 50, Galea, L. M., Dalton, S. M. C., Beck, E. J., Cashman, C. J., and Probst, Y. C., Update of a database for estimation of wholegrain content of foods in Australia, 23–29, Copyright (2016) with permission from Elsevier.

**Table 1 ijerph-17-09229-t001:** Participant characteristics.

	Participants *N* = 828		Participants with Plausible Energy Intake *n* = 703	
Wholegrain intake (g/day)	Mean (±SD)	95% CI	Mean (±SD)	95% CI
	19.5 (±14)	18.5–20.4	18.5 (±13)	17.6–19.5
	Median	IQR	Median	IQR
	18.0	8.3–27.4	17.5	7.9–26.0
	*N*	%	*n*	%
Maternal characteristics				
Maternal age at baseline (years)				
<25	74	8.9	63	9.0
25–34	577	69.7	485	69.0
≥35	177	21.4	155	22.0
Number of children				
1	389	48.6	337	49.5
2	291	36.3	247	36.3
≥3	121	15.1	97	14.2
Maternal education				
High school/vocational	356	43.2	311	44.4
Some university and above	468	56.8	389	55.6
IRSAD				
Decile 1–2: Most disadvantaged	120	14.6	102	14.6
Decile 3–4	173	21.0	146	20.9
Decile 5–6	174	21.2	150	21.5
Decile 7–8	160	19.5	136	19.5
Decile 9–10: Most advantaged	195	23.7	163	23.4
Maternal country of birth				
Australia or New Zealand	610	74.1	511	73.2
United Kingdom	31	3.8	24	3.4
India	50	6.1	43	6.2
China	37	4.5	35	5.0
Asia—other ^a^	52	6.3	47	6.7
Other ^a^	43	5.2	38	5.4
Child characteristics				
Sex				
Male	452	54.6	389	55.3
Female	376	45.4	314	44.7
Age of solid food introduction				
<17 weeks	202	24.6	1630	23.4
≥17 weeks	620	75.4	534	76.6

^a^ No single country within this group had sufficient number of participants to warrant separate statistical analyses. SD: Standard Deviation; IQR: Interquartile Range; CI: Confidence Interval IRSAD: Index of Relative Socio‑Economic Advantage and Disadvantage, where 1 = most disadvantaged and 10 = most advantaged.

**Table 2 ijerph-17-09229-t002:** Food group contribution to usual wholegrain intake for all children (*n* = 828) and for food group consumers only.

Food Group	All Participants			Consumers Only		
	Mean (SD) (g/Day)	95% CI^2^	% contribution of Food Group to Wholegrain Intake ^a^	*n* (%)	Mean (SD) (g/Day)	95% CI
Total Cereals and cereal products ^b^	52.5 (46.6)	49.3–55.7	89.90%	710 (85.7)	61.2 (44.7)	57.9–64.5
Flours and other cereal grains and starches ^c^	5.5 (17.0)	4.35–6.67	9.40%	130 (15.7)	35.1 (28.3)	30.2–40.0
Regular breads, and bread rolls (plain/unfilled/untopped varieties)	15.5 (25.7)	13.8–17.3	26.60%	374 (45.2)	34.4 (28.6)	31.5–37.3
English‑style muffins, flat breads, and savory and sweet breads	1.6 (11.1)	0.9–2.4	2.80%	45 (5.4)	30.0 (38.2)	18.6–41.5
Pasta and pasta products (without sauce)	0.9 (6.0)	0.4–1.3	1.50%	46 (5.6)	15.4 (20.8)	9.18–21.5
Breakfast cereals, ready to eat	23.4 (29.0)	21.4–25.4	40.00%	472 (57.0)	41.0 (27.4)	38.5–43.5
Breakfast cereals, hot porridge style	4.2 (14.9)	3.2–5.2	7.20%	105 (12.7)	32.9 (28.4)	27.4–38.4
Savory biscuits	0.8 (3.4)	0.6–1.0	1.40%	82 (9.9)	8.5 (7.4)	6.9–10.1
Dairy milk substitutes, unflavored ^d^	0.8 (9.9)	0.1–1.5	1.40%	16 (1.9)	41.1 (60.1)	9.1–73.2
Total Commercial Infant foods	4.9 (11.3)	4.1–5.6	8.30%	390 (47.1)	10.3 (14.7)	8.8–11.8
Infant cereal products ^e^	3.2 (10.7)	2.5–3.9	5.50%	170 (20.5)	15.6 (19.1)	12.7–18.5
Infant foods—other ^f^	1.7 (3.4)	1.4–1.9	2.80%	302 (36.5)	4.5 (4.4)	4.0–5.0

^a^ Food groups providing < 1% contribution to wholegrain intake are not listed, ^b^ Cereal and cereal products is inclusive of cereal-based products and dishes. ^c^ Flours and other cereal grains and starches providing wholegrains included cooked grains such as rice, quinoa, and buckwheat; oats other than those in porridge or muesli; and some specialty breads such as rye and spelt. ^d^ Unflavored dairy milk substitutes providing wholegrains included rice milk and oat milk. ^e^ Infant cereal products providing wholegrains included infant cereals in both dry and ready‑to‑eat forms. ^f^ Infant foods‑other providing wholegrains included rice cakes, biscuits and oat or muesli bars marketed as infant and/or toddler products. 95% CI: 95% confidence interval.

**Table 3 ijerph-17-09229-t003:** Factors independently a associated with usual wholegrain intakes of children aged 12–14 months (unadjusted and adjusted mean values and 95% confidence interval) (*n* = 828).

	Unadjusted Mean		Adjusted Mean				
	g/day	95% CI	g/day	95% CI	β	SE	*p*
Maternal Characteristics							
Maternal age at baseline (years)							
<25	14.8	12.1–17.4	10.3	6.4–14.1	−5.468	2.118	0.01
25–34	20.1	18.9–21.3	16.2	14.2–18.2	0.467	1.236	0.706
≥35	19.3	17.4–21.1	15.7	13.2–18.3	REF		
Maternal education							
High school/vocational	18.9	17.5–20.3	13.1	10.6–15.6	−1.908	1.082	0.078
Some university and above	19.8	18.5–21.1	15	12.8–17.2	REF		
IRSAD							
Decile 1–2: Most disadvantaged	18.8	16.2–21.3	14.4	11.2–17.5	−0/071	1.662	0.966
Decile 3–4	19.2	17.3–21.2	13.7	11.0–16.5	−0.709	1.476	0.631
Decile 5–6	20.3	18.0–22.5	15.1	12.4–18.0	0.724	1.454	0.619
Decile 7–8	17.7	15.8–19.6	12.6	9.7–15.4	−1.843	1.498	0.219
Decile 9–10: Most advantaged	20.6	18.5–22.8	14.4	11.7–17.2	REF		
Maternal country of birth							
Australia or New Zealand	20.7	19.6–21.8	18.6	REF	REF		
United Kingdom	23.1	17.3–28.9	20.6	15.5–25.6	1.977	2.648	0.456
India	19.2	15.4–23.0	15.7	11.5–19.9	−2.880	2.104	0.172
China	9.7	6.8–12.6	6.1	1.3–11.0	−12.425	2.405	<0.001
Asia—other ^b^	12.3	9.5–15.2	9.4	5.4–13.5	−9.132	2.074	<0.001
Other ^b^	17.1	13.1–21.1	14	9.5–18.4	−4.601	2.209	0.038
Number of children							
1	19.7	18.3–21.2	15.3	13.1–17.4	REF		
2	20.1	18.5–21.7	15.2	12.8–17.7	−0.074	1.093	0.946
≥3	17.4	15.2–19.7	11.7	8.6–14.8	−3.587	1.479	0.016
Child Characteristics							
Sex							
Male	20.1	18.8–21.4	15.2	12.9–17.4	2.191	0.973	0.025
Female	18.7	17.3–20.0	13	10.6–15.3	REF		
Age of complementary food introduction							
<17 weeks							
≥17 weeks	19.2	17.2–21.3	14.2	11.6–16.7	0.184	1.164	0.874
	19.5	18.5–20.6	14	11.8–16.1	REF		

a Multivariable linear regression adjusted for age of child at time of 24‑h recall in addition to all maternal and child characteristics in the table. b No single country within this group included sufficient sample size to warrant separate statistical analyses. IRSAD, Index of Relative Socio‑Economic Advantage and Disadvantage, where 1 = most disadvantaged and 10 = most advantaged.

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
