# Peer review of "Sources and Determinants of Wholegrain Intake in a Cohort of Australian Children Aged 12–14 Months"

_ijerph, 2020, doi:10.3390/ijerph17249229_

Round 1

Reviewer 1 Report

The study from ijerph-1014237, titled “Sources and determinants of wholegrain intake in a cohort of Australian children aged 12-14 months” presents good results on the proposed topic and could be interesting for the audience of IJERPH.

Author Response

Thank you for your comment. Revisions have been made in response to reviewers 2-4.

Reviewer 2 Report

Thank you for the opportunity to review this paper. This manuscript aimed to determine the intake and food sources of whole grains in a cohort of 828 Australian children aged 12-14 months which is of great significance. However, the paper is not written to a high standard and needs considerable improvement.

  1. This study used General Linear Model (GLM) to analyze the influencing factors of infants' whole grain intake. I think it is unreasonable since there are many influencing factors of infants' whole grain intake. Instead, multiple linear stepwise regression analysis should be used to analyze the influencing factors of infants' whole grain intake more rigorously.
  2. Regression coefficient, standard error, and the standardized regression coefficient should be provided in Table 3.
  3. With rapid social and economic development, eating habits and concepts have undergone great changes. This study was conducted in 2013 which is 7 years ago. Given the facts, this can be presented as a limitation.

Author Response

Thank you for your comments. We have made revisions in the revised manuscript, in response to your feedback as follows:

  1. We defend the use of the multivariable general linear model to rigorously analyse the influencing factors of young children’s wholegrain intake. We simultaneously adjusted for a small number of a priori selected socio-demographic variables that have been reported in the literature to be associated with a variety of dietary outcomes related to young children, for which we had data.  Thus, a step-wise approach is unnecessary.

    While we described the use of multivariable GLM in the methods section we acknowledge that this may not have been readily apparent in the way in which we reported the results.  We trust that the changes that we have made to both the text (line 121 on) and table 3, now makes it clear that we adjusted for known socio-demographic determinants of children’s dietary intake.

  2. Upon reviewing the data the outcome variable was slightly positively skewed but the non-normality of the outcome was considered to be inconsequential. Of greater concern is the normality of the standardised residuals, which was found to be acceptable, therefore we reran the analysis on the original non-transformed data. Table 3 has been revised and the results are presented as the unadjusted and adjusted mean usual wholegrain intake, with 95% CI and the regression coefficient, standard error and p values obtained from regression analyses, as requeseted.

  3. This has been added to the limitations at line 272

Please note the line references in this response reflect the track changes version of the document. Thank you.

Reviewer 3 Report

The paper is prepared in clear, logical and comprehensible way. The research scope incorporates in relatively narrow range of research focusing on diet and health problems of children in developed countries.

My comments/suggestion to improve papers are as follows:

  • Abstract may be extended by research method you applied.
  • In Introduction I suggest to underline your motivations to do such research. I suggest also to present SMILE project and its results in a little wider way, while there are some published papers related to this project and its results (for example, Coxon et al . 2019, Devenish et al. 2019, Scott et al. 2016….).
  • I am confused about “12-14 Months” not “12 Months” or “8-14 Months” in title, as you do research when children were 3, 6, and 12 months of age (line 61). By the way, why not 3, 6, 12 and 24 like in other papers related to the SMILE effects (for example Coxon at al. 2020).
  • Explain IRSAD a little bit or give some references.
  • In table 1 there are “Participants with plausible energy…” and in table 2 there are “Consumers only”. Why did you specify that groups? What for? What are the reasons behind this? It is not clear and there are no comments and discussion considering that groups. They look like “table fillers”.
  • Referring to your recommendations; how to solve problem “wholegrain versus sugar in food”? I suggest also to refer your recommendations to groups like, for example, parents, food producers, governments, schools, etc…, as their possibilities and range of activities are limited and separated to some extent.

In Conclusion underline your contribution, refer to so far research and indicate further research directions.

Author Response

The authors thank you for your comments. We have revised the manuscript in response to your feedback. A point-by-point response to your comments is attached.

Reviewer 4 Report

The work is attractive to the  readers, interesting, correct. It's well structured. It's clear and concise. It addresses very well the processing of a large number of data. I think it has the conditions to be admitted to publication in this journal making some small modifications that can improve it.

The first and most important is to change Figure 1 to a larger one, so that the text can be read easily.

The second is to incorporate charts that help to more simply visualize the contents of the tables.

The summary is adequate. The introduction perfectly justifies the state of the art and the interest of research. The methodology is very well explained, replicable, and contains all the essential Ethic requirements in such a scientific study. The statistic is adequate.

The discussion leads to expected conclusions, which are consistent with common sense, but scientific rigour in the field of social sciences is precisely that: to demonstrate numerical evidence, or to raise contrary hypotheses

We congratulate the authors on the design and execution of this work and understand that it must be completed with others of this line of research, where it further delves into the correlation of children's food and the effects of globalization on foodguidelines.

Author Response

Thank you for your comments. We have made revisions to the manuscript in response to your feedback as follows:

A high resolution copy of the figure was uploaded separately but has now also been inserted here with the resolution retained and the size increased. This figure has been moved to the methods section (line 88) as requested by reviewer 5.

While we agree that visualisation of data can be useful, it is not customary with data of this type. We believe that for this topic there is little that would be suited to presentation via a chart and we are unsure to which particular results you are referring. If you think it is necessary to include a chart, please could you identify which data should be represented in chart form, and give us an example of the chart you have in mind. Thank you.

Please note the line reference in this response reflects the track changes version of the document

The authors appreciate your time.

Reviewer 5 Report

Overall an article entitled “Sources and Determinants of Wholegrain Intake in a Cohort of Australian Children Aged 12-14 Months” is written very well, and the discussed topic is of great importance for the nutrition of young children. I have only a few comments regarding methodology in particular:

You use references to article 19 (Devenish, G.; Ytterstad, E.; Begley, A.; Do, L.; Scott, J. Intake, sources, and determinants of free sugars intake in Australian children aged 12-14 months. Matern. Child Nutr. 2019, 15, e12692, doi:10.1111/mcn.12692),  where this methodology is clearly written both in terms of the characteristics and the research carried out. So please describe the methodology in more detail in this article.

So please add:

Briefly, 2,147 mothers and 2,181 newborns, including 34 pairs of twins, were recruited from the three major maternity hospitals in Adelaide from July 2013 until August 2014.

The Southern Adelaide Clinical Human Research Ethics Committee approved the study (HREC/50.13, approval date: 28 Feb 2013) as did the South Australian Women and Children Health Network (HREC/13/WCHN/69, approval date: Aug 7, 2013).

Participants were invited to complete questionnaires at recruitment, and when their child reached 3, 6, 12, and 24 months of age. These questionnaires collected information on dental and dietary behaviours and were available in paper, electronic, and telephone interview forms

move Annex A to the methodology

Maybe it is worth summarizing the conclusions by key messages

Author Response

Thank you for your comments. We have made revisions to the manuscript in response to your feedback as follows:

These components of the methodology are all described in methods section 2.1, with the recommended inclusions commencing at line 58 (“…data from the SMILE birth cohort, consisting of 2181 children born to 2147 mothers…”), line 70 (“This study was approved by the Southern Adelaide Clinical Human Research Ethics Committee …”) and line 66 (“Socio demographic characteristics were collected via a paper-based questionnaire at recruitment, and the participants’ choice of postal, online or telephone administered questionnaire when the children were 3, 6, and 12 months of age…”). These have been revised for clarity, and further information added to the methods at lines 115 and 121 in response to methodological details requested by other reviewers.

The Appendix figure has been moved to line 88 and changed to Figure 1. A larger version has been included as requested by reviewer 4.

The conclusion has been rewritten to summarise key messages and respond to additional recommendations from reviewer 3. Thank you for your comments

Please note the line references in this response reflect the track changes version of the document

The authors appreciate your time.

Round 2

Reviewer 2 Report

This manuscript aimed to determine the intake and food sources of whole grains in a cohort of 828 Australian children aged 12-14 months which is of great significance.The author has made modifications according to the modification requirements.I think it can be accepted and published